# Work Organization Factors Associated with Health and Work Outcomes among Apprentice Construction Workers: Comparison between the Residential and Commercial Sectors

**DOI:** 10.3390/ijerph18178899

**Published:** 2021-08-24

**Authors:** Ann Marie Dale, Diane S. Rohlman, Lisa Hayibor, Bradley A. Evanoff

**Affiliations:** 1Division of General Medical Sciences, Washington University School of Medicine in St. Louis, 4523 Clayton Avenue, CB 8005, St. Louis, MO 63110, USA; hayibor@wustl.edu (L.H.); bevanoff@wustl.edu (B.A.E.); 2Healthier Workforce Center of the Midwest, Iowa City, IA 52242, USA; diane-rohlman@uiowa.edu; 3Department of Occupational and Environmental Health, College of Public Health, University of Iowa, 145 N. Riverside Drive, Iowa City, IA 52242, USA

**Keywords:** worker injury, workplace health supports, psychosocial job factors, nontraditional workplace hazards, Total Worker Health

## Abstract

There are substantial differences in work organization between residential and commercial construction sectors. This paper examined differences in work factors between construction sectors and examined the association between sector and health behaviors, health outcomes, and work outcomes. We surveyed 929 male construction apprentices (44% residential and 56% commercial) and found that residential apprentices reported fewer workplace safety policies, higher frequency of heavy lifting, and greater likelihood of reporting musculoskeletal pain compared to apprentices in commercial work. Residential apprentices reported higher job strain, lower supervisor support, more lost workdays due to pain or injury, and lower productivity related to health than commercial apprentices. Multivariate Poisson regression models controlling for multiple work factors showed that residential construction work, high job strain, heavy lifting, low coworker support, and low supervisor support were each independently associated with one or more work or health outcomes. These findings suggest that interventions should seek to improve coworker and supervisory supportive behaviors, decrease job strain, and reduce organizational stressors, such as mandatory overtime work. Our study shows disparities in health and safety between construction sectors and highlights the need for interventions tailored to the residential sector.

## 1. Introduction

Work organization has been associated with poor health outcomes in several industries. How the work is coordinated and controlled to accomplish the goals of the company may create physical and psychosocial burdens on the workforce. Common factors that lead to poor health include rotating or long shift work, seasonal jobs, mandatory overtime, and jobs with low autonomy [1,2]. These factors are well-known in many industries and include rotating schedules for air traffic controllers [3], long shifts for healthcare [4], mandatory overtime for critical care and public service providers [5,6], and low job autonomy in domestic and food service workers [7,8]. Construction operations incorporate many work organization factors and restrictions on workers that have previously been associated with negative health outcomes [9,10]. However, the construction industry has other factors that contribute to worker health concerns.

Construction is among the most hazardous industries, with high rates of worker fatalities and non-fatal injuries. In addition to these well-known rates of work injury, many construction workers suffer or experience chronic diseases, are more likely to receive medical care for musculoskeletal conditions, and have a greater prevalence of functional health problems that limit work when compared to other working populations [11,12,13]. Construction workers have higher rates of alcohol use, smoking, and poor diet when compared to the working population as a whole [14,15,16,17,18]. These disparities in health-related behaviors contribute to the higher rates of mortality, morbidity, and health-related work disability seen among construction workers. While these health behaviors have traditionally been considered unrelated to work and outside the scope of workplace safety and health programs, there is growing evidence that these health behaviors are driven in part by modifiable work organization and work environment factors [19]. We previously studied a cohort of apprentice carpenters and floor layers to examine associations between work organization and environment factors on work and health outcomes of relevance to employers, including missed work due to work-related injury, missed work due to any pain or injury, health-related work ability and productivity, and use of prescription medications for pain [9]. This study found associations between these outcomes and multiple work factors, including job strain, safety behaviors of coworkers, and overtime policies, suggesting that work organization and environment factors influence health and work outcomes among young construction trade workers. Past studies have shown that young construction workers are at greater risk of injuries; however, there are opportunities to intervene through education on modifiable risk factors during the apprentice training [11,20]. In this study, we extended our work to examine differences in work organization and environment factors and their associations with health behaviors, health outcomes, and work outcomes between two different construction sectors: commercial construction and residential construction.

Work organization and environment differs greatly between commercial and residential construction. Residential construction is typically characterized by fewer safety resources; smaller employers with less formal organization and project oversight; frequently changing work environment, higher turnover of workers on projects; fewer safety regulations; less training; and small, often scattered, crews with less knowledgeable leaders [13,16,21]. The risk of fatal and non-fatal injury is highest among small construction employers, “who are less likely to embrace essential safety culture practices and are slow to adopt new approaches to occupational safety and health” [22]. Poorer safety climate has been reported on residential construction sites [23]. While differences in work-related injuries and specific safety practices between the residential and commercial construction sectors have been described, this study sought to assess differences in broader health policies and practices, and their potential effects on health behaviors and health and work outcomes.

We based this study on a conceptual model of Total Worker Health used in our prior study among apprentice carpenters [9]. Our conceptual model (Figure 1) shows how a variety of work factors can directly or indirectly influence important health behaviors, health outcomes, and work outcomes. The purpose of the current project was to compare workplace risks and supports for safety and health between residential and commercial apprentice construction workers and their associations with three types of outcomes: health behaviors, health outcomes, and work outcomes. We hypothesized that apprentices employed by residential contractors would report fewer supports for safety and health, and would report poorer health behaviors, health outcomes, and work outcomes than apprentices in commercial work. We also hypothesized that differences in work organization between the residential and commercial construction sectors would be associated with poorer health and work outcomes among residential apprentices.

## 2. Materials and Methods

### 2.1. Participants

Apprentices attending training classes offered by two union apprenticeship programs in Missouri (Carpentry and Floor Laying) at any time between 15 February and 14 June 2017 were invited to complete a survey on health and work. Each apprentice received a survey packet containing the study description and invitation to complete the survey, and the survey at the beginning of the school day, and were provided time to complete the survey. Participants provided informed consent by completing the survey and returning it to a research technician who was located at the school. Participants were compensated $15 for survey completion and were only allowed to complete the survey once. The study was conducted according to the guidelines of the Declaration of Helsinki, and approved by the Institutional Review Board (or Ethics Committee) of University of Iowa (protocol code 201611819 and approved on 23 January 2017)

### 2.2. Survey

The survey contained 78 items addressing four major domains that have previously shown associations between work and health: work organization and environment, health behaviors, health outcomes, and work outcomes. Many studies have shown negative work-related health consequences from organizational and environmental factors such as unhealthy physical work environment and low autonomy and poor managerial support [24]. There are many associations between health behaviors such as smoking, poor diet, and lack of physical activity and poor health outcomes. We selected items from each domain which are more prevalent among construction workers [14,25,26]. Items, definitions, and sources for each of these domains are shown in Table 1. We collected demographic information including age, sex, years in trade, apprentice training term, and type of work. Apprentice carpenters and floor layers were classified as working in residential or commercial construction by the question, “What type of construction is your current (or most recent) project?” Note that each apprentice signs a letter of agreement to work for a signatory contractor. These contractors typically perform either residential construction (home building) or commercial construction. Work organization and environment factors included work hours, policies, perceived physical and psychosocial job demands, job security, safety climate, social support, and commute time to work. Health behaviors included alcohol consumption and use of tobacco products. Health outcomes included musculoskeletal symptoms of the neck, hand/wrist, lower back, and knee; physician visit in the past 12 months for musculoskeletal symptoms; use of prescription drugs for pain; physical or mental health assessed by the SF-8 [27]; and fatigue. Work outcomes included work-related injuries, missed work due to pain or injury, health-related work ability, and health-related work productivity.

### 2.3. Analysis

We calculated the strain ratio from reported job demands and decision latitude on the Job Content Questionnaire (JCQ) [29], and then we computed the scores of scales for Zohar’s Safety Climate [18] and SF-8 [16]. To contrast the proportions or means of outcomes between apprentices working in residential and commercial construction, we used *t*-tests, chi square tests, and Wilcoxon rank sum tests (for continuous, dichotomous, and ordinal variables, respectively). We conducted univariate and multivariate Poisson regression with robust sandwich estimators to determine the relative risk of each independent variable to one or more outcomes [36]. We selected a group of ten work and health outcomes (missed work due to work-related injury, missed work due to any injury or pain, work ability, health-related productivity, lower back pain, use of prescription medications for pain, poor mental health, fatigue after work, frequency of alcohol consumption, and current smoking status). In conducting these analyses, we selected independent variables a priori, with ten work organizational factors for each multivariate analysis: mandatory overtime, smoking restriction policy, hearing protection policy, glove requirement policy, frequent lifting of heavy loads, high job strain, low coworker support from the JCQ, low supervisory support from the JCQ, supervisor support for safety, and coworker support for safety [29,30,37].

## 3. Results

### 3.1. Study Population and Demographics

We distributed 1070 surveys to construction apprentices; 963 were completed and returned (response rate = 90%). As only 3.5% of respondents were women, we limited our analyses to male workers (*n* = 929). There was no meaningful difference in age (27.8 years overall), years in trade (2.6 years overall), or apprentice training term (59.3% beyond first year of training) between residential and commercial workers. Those working in residential construction were more likely (*p* < 0.05) to be white (90.4% vs. 85.5%), less likely to be obese (15.7% vs. 23.3%), and less likely to be eligible for union benefits based on hours worked (79.1% vs. 84.7%) than those working in commercial construction. A total of 408 apprentices (44%) were currently or most recently employed on residential construction projects, and 56% on commercial construction.

### 3.2. Work Organization and Environment Factors

We examined worker-reported workplace safety policies and supports, workplace psychosocial factors and job demands, and work factors related to individual health behaviors.

For workplace safety policies and supports, commercial workers were more likely to report that their employer had policies on required use of hearing protection and on the use of respirators, gloves, high visibility clothing, ventilation, and protective clothing (data shown for hearing protection and glove use). There were no differences in requirements for hard hats and safety glasses. Commercial workers more often reported access to water at the worksite and access to seasonal warming or cooling. There was no reported difference in employer policies on drug and alcohol testing; however, hearing tests were more likely to be required in commercial construction (data not shown). Residential workers were more likely to report frequent heavy lifting at work.

Examination of psychosocial factors and job demands found that residential apprentices reported higher levels of job strain, related to their higher reported job demands and lower reported job skill discretion (job control) on the Job Content Questionnaire (JCQ). JCQ scores for social support from supervisors and coworkers were also lower in residential construction, as were scores on scales for supervisory support for safety and for coworker safety [30,37]. Residential workers also reported lower job satisfaction. Some aspects of work organization were better on residential projects—mandatory overtime was less common and worker commutes were shorter. Overall, 22.7% of workers reported precarious work (poor job security or job instability), with no difference between sectors.

Regarding workplace supports for individual health behaviors, workplace restrictions on smoking were less common on residential projects. Workers in residential construction also reported lower availability of sunscreen, hand sanitizer or a place to wash hands, access to food for purchase, a place to refrigerate or store food, or a designated eating area (data not shown).

### 3.3. Health Behaviors and Health Outcomes

Residential workers reported using alcohol more days per month and binge drinking (five or more drinks on one occasion) more frequently than commercial workers. Residential workers were slightly more likely to report current smoking, though not e-cigarette use.

Health outcomes were poorer among residential apprentices when compared to commercial. Residential workers were more likely to report pain or discomfort in the past year in all body parts assessed: hands, neck, lower back, and knees. Residential workers were also more likely to have seen a physician in the past year for musculoskeletal pain (25.6% versus 19.3%) and more likely to have received a prescription pain medication. These workers also reported poorer mental health and physical health than commercial workers on the SF-8 scales, and they more often reported fatigue and low levels of energy.

As shown in Table 2, these young construction workers reported a very high overall rate (9.5%) of having missed work in the past year due to a work-related injury, with a higher proportion among residential workers (13.5% vs. 6.4%). Residential workers more frequently reported missing days of work due to work-related pain or injury and missing days due to any pain or injury; however, there was no difference in missed days due to illness. Residential workers were more likely to report both decreased work productivity and decreased work ability with respect to health.

Prior to regression analysis, we removed the hearing protection policy due to its high correlation with glove requirement policy, and supervisor and coworker support for safety due to high correlations with supervisor and coworker support on the JCQ scale. In univariate Poisson regression models (Table 3), residential construction was significantly associated with the outcomes of missed work due to work-related injury, missed days due to any injury or pain, health-related productivity, low back pain, prescribed medication for pain, poor mental health, being tired after work, and frequency of alcohol consumption. There were also associations between residential construction and the outcomes of work ability and cigarette smoking that did not achieve statistical significance (lower limit of confidence interval was 0.99). Three of seven work organization factors (mandatory overtime, smoking restriction policy, and glove requirement policy) were not significantly associated with any outcomes in univariate analysis; the other four factors (frequent heavy lifting, high job strain, low coworker support, and low supervisory support) were associated with the majority of health and work outcomes. After controlling for these seven other work factors in multivariate analyses, residential construction was independently associated with missed work due to work-related injury, health-related productivity, low back pain, and being tired after work (models shown in Table 3). High job strain was independently associated with six of ten outcomes in the multivariate models (missed days due to work-related injury, missed days due to any injury or pain, work ability, prescribed pain medication, poor mental health, and feeling tired after work) (data not shown for all models). Heavy lifting, low coworker support, and low supervisor support were each associated independently with one or more outcomes.

## 4. Discussion

Our study found high rates of unhealthy behaviors and evidence of poor health and work outcomes among apprentice construction workers as a whole. There were large differences in health behaviors and outcomes between young workers in the residential and the commercial construction sectors, and large differences in the availability of some workplace organizational and environmental supports for health between sectors. Safety practices and injury rates have been found to be worse in residential construction, particularly in small residential contractors, than in commercial construction [13,21,22]. Our results show that health behaviors, health outcomes, and health-related productivity are also worse in the residential sector. Residential construction was an independent predictor of multiple work and health outcomes in multivariate models adjusted for important work environment and organization factors that differ between construction sectors. Some, but not all, of the differences seen between sectors were explained when adjusting for factors, including safety requirements, lifting heavy loads, job strain, and coworker/supervisor support.

Our study found few demographic differences between apprentice carpenters working residential or commercial construction. The smaller proportion of residential workers who were eligible for union benefits (based on hours worked) may reflect the more seasonal and cyclical nature of the residential home building environment; however, we did not see differences in reported job stability and job security between the two sectors.

The finding of a higher rate of work-related injuries and lost days among residential construction has been reported in previous studies [38,39]. In our study, residential workers more frequently reported missing days of work due to a work-related injury or to any injury; however, there was no difference in missed days due to illness between sectors. This finding suggests specificity of response given the higher prevalence of musculoskeletal symptoms, medical use, prescription pain medication use, and heavy lifting seen among residential workers. Our study deliberately asked about lost days due to all injuries and pain, as well as injuries attributed to work, due to the well-described disincentives to reporting work-related injuries and illnesses in construction [40].

Unique findings from this study include the increased likelihood of residential workers to report lower work productivity due to health, and lower work ability with respect to the physical demands of work, though not reduced work ability related to mental demands of work. These findings may reflect both the higher prevalence of musculoskeletal pain and the higher workplace physical demands reported by residential workers and are supported by national data showing higher rates of overexertion injuries among workers in the residential sector [16]. The higher rates of lost work due to injury and decreased work ability and productivity among residential apprentices are mirrored in their poorer health outcomes and higher rates of musculoskeletal symptoms, in their increased likelihood of having seen a physician in the past year for musculoskeletal pain, and in their increased likelihood of receiving a prescription pain medication. The links between work injuries and physical demands with the use of opioids have become increasingly apparent in construction workers [41,42], and those in similarly demanding industries, such as mining and commercial fishing.

The high prevalence of musculoskeletal disorders (MSDs), particularly among residential workers, is an example of how the absence of effective workplace policies, programs, and practices affect construction worker health and well-being. The physically demanding nature of the work, including manual materials handling, awkward and static postures, vibration, and strenuous physical exertions, helps explain why MSDs account for the majority of all injuries resulting in days away from work [43]. Complaints of fatigue and low energy after work are likely related to the physical demands of this work as well. Despite the existence of practical, low-cost solutions for reducing physical exposures, and efforts by NIOSH and other groups to disseminate these solutions [16,44], prevention efforts to reduce MSDs in construction have not been systematically incorporated in most safety programs [45]. In our study, rates of MSD symptoms and seeking care from a physician for MSD were high in both construction sectors, but markedly higher in residential construction, where workers also reported more frequent lifting of heavy loads.

More frequent alcohol use, including more frequent binge drinking, was reported by residential construction workers. Higher rates of alcohol use among construction workers than the general population have been previously reported [14,18]; however, to our knowledge, this is the first report of differences in alcohol use reported between different construction sectors within the same trade. Alcohol use among construction workers has received much less attention than smoking and opioid use. The overall rates of smoking and e-cigarette use were quite high in both construction sectors; although not significant, residential workers had higher rates of current smoking than commercial workers. This is consistent with other studies which have found that smoking is generally tolerated on construction worksites and that construction workers have one of the highest rates of smoking and among the lowest rates of workplace policies restricting smoking [46,47]. While workplace smoking policies have been associated with higher quit rates and lower rates of smoking in other studies, our study did not find that reported workplace smoking policies were significantly associated with current smoking status [48,49]. Unlike the MassBUILT study, we also did not find that contractor safety climate was associated with smoking rates [50].

Consistent with other studies, differences in workplace safety policies and supports were apparent between the two sectors, with commercial employers being more likely to require use of personal protective equipment and to require surveillance examinations (hearing and respiratory testing), as well as to provide water and seasonal warmth or cooling. Our study found several differences in psychosocial factors and job demands between commercial and residential workers. Residential apprentices reported lower scores for supervisor and coworker safety; these and other measures of safety climate have been linked to workplace safety practices and also to greater fatigue [21,23], which was also reported by residential apprentices. Residential apprentices reported higher job demands and lower job skill discretion than commercial workers, resulting in a higher calculated job strain that was independently associated with multiple health and work outcomes. Workplace supports for individual health behaviors (those not directly related to work activities) were more often present in commercial construction. Workplace restrictions on smoking were less common on residential projects, as was access to food for purchase, a place to refrigerate or store food, or a designated place to eat.

Differences in safety practices between commercial and residential construction have long been recognized. Safety training practices, the quality of safety programs, and use of protective equipment for common hazards have been noted to be lower in residential than commercial construction [13,21]. In addition, residential contractors are more commonly non-union, so their work force has generally received less safety training than these union workers [51]. Most residential contractors are small, and the risk of fatal and non-fatal injuries is highest among small construction employers, who are less likely to embrace essential safety culture practices, are slow to adopt new approaches to occupational safety and health, and lag far behind in terms of adopting safety cultures and management practices [22]. Our study findings show disparities in work and health outcomes within an already hazardous industry and highlight the need for interventions to target the high rates of poor work and health outcomes and relatively low rates of workplace supports for health seen in both construction sectors, but particularly in residential construction.

Our study had a number of limitations. The cross-sectional design with self-report of both work conditions and health and work outcomes should engender caution in making any causal attributions. Annual follow-up of this worker group is continuing and will allow for future longitudinal analyses. Our classification of workers in residential or commercial construction is based on self-report of the current or most recently held job. Since some workers move between residential and commercial projects, the actual differences in outcomes and exposures may in reality be larger than those found in this study. Our study was limited to union apprentice workers in one construction trade in one metropolitan area; results may have been different in different trades or locations. Results may also differ in a non-unionized workforce, as well as with general construction workers whose behaviors and health outcomes may change with age and experience.

## 5. Conclusions

Our study findings show disparities in health and safety between the residential and commercial construction sectors and highlight the need for interventions to target the high rates of poor work and health outcomes seen in both sectors. The differences between the sectors is due in part to the nature of the work, with smaller crews, less oversight, short-term projects, and less formal work conditions common to residential work. As a whole, the construction apprentice workforce is characterized by low wages, job insecurity, and contract work/temporary employment, issues faced by a growing number of industries due to changes in work organization [52]. The changing nature of work in other companies, resembling aspects of the complex construction organization, will likely create poorer worker health. Despite these differences, there are opportunities for employers to implement formal workplace policies and practices to improve worker health and safety in residential construction. Residential contractors may be encouraged by the improved level of health and safety realized by the consistent application of policies across all projects. However, even more compelling would be organizational policies and programs that go beyond traditional safety hazards and strive to improve the overall health of the construction worker.

## Figures and Tables

**Figure 1 ijerph-18-08899-f001:**
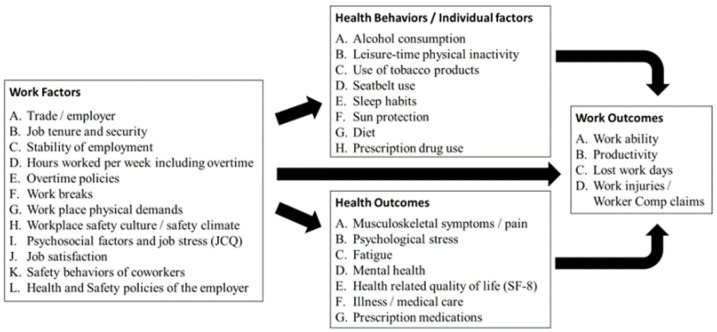
Conceptual model for survey variables.

**Table 1 ijerph-18-08899-t001:** Definitions and sources of work outcome, health outcome, health behavior, and work factor variables.

Variable	Definition/Source
*Work organization and environment*	
Residential construction work	Yes/No—versus commercial construction work
Policies for (a) hearing protection, (b) gloves, (c) water access, or (d) seasonal temps	Yes/No
Frequent heavy load lifting	Lift heavy loads “often” or “always” [28]
High job strain	Strain ratio > 1 [29]
Low (a) supervisor and (b) coworker support	Below median [29]
Safety Climate scale score (a) Supervisor and (b) coworker support	Range 0–100 [30]
Low job satisfaction	Response other than “very satisfied” on 4-point scale [31]
Commute time (minutes)	Duration of commute, minutes each way
Precarious work	Disagree with “my job security is good” OR work not “regular and steady” [29]
Poor job security	Disagree with “my job security is good” [29]
Job instability	Report work not “regular and steady” [29]
(a) Mandatory overtime, (b) smoking restriction policy, (c) access to food, or (d) designated eating area	Yes/No
*Health behaviors*	
Alcohol consumption days/month	Days had at least 1 drink, past month [32]
Binge drinking days/month	Days had 5 or more drinks on one occasion, past month [32]
Current cigarette smoker	Smoke cigarettes “everyday” [32]
Current e-cigarette user	Use electronic cigarettes “everyday” or “some days [32]
*Health outcomes*	
Pain/discomfort of (a) neck/shoulder, (b) hand/wrist, (c) lower back, and (d) knee	Any, past 12 months [33]
Doctor visit due to musculoskeletal symptoms	For neck/hands/lower back/knees, past 12 months [33]
Prescribed medication for pain	Any, past 12 months
Poor (a) physical and (b) mental health	Physical or mental SF-8 scale score below 1st quartile [27]
Low energy (past 4 weeks)	“Some”, “a little” or “none” [27]
Tired after work	Tired “often” or “very often” after work [28]
*Work outcomes*	
Missed days due to (a) work-related injury, (b) any injury or pain, or (c) any illness	1 or more missed days, past 12 months
Poor work ability	Score below 9 on 10 point scale [34]
Poor work ability for (a) physical demands and (b) mental demands	“Poor” or “rather poor” on 5 point scale [34]
Poor health related productivity	Score above 1 on 10 point scale [35]

**Table 2 ijerph-18-08899-t002:** Prevalence and means of work outcomes, health outcomes, health behaviors, and work factors between residential and commercial workers.

Variable	Total *n* = 929	Commercial *n* = 520 ^a^	Residential *n* = 408 ^a^	*p*-Value ^b^
*Work organization and environment*	%			
Residential construction work	44.0	-	-	-
Hearing protection policy	51.0	63.8	34.6	<0.001
Glove requirement policy	46.2	65.0	22.3	<0.001
Water access	49.9	62.7	33.6	<0.001
Seasonal warmth and cooling	33.3	50.0	12.0	<0.001
Frequent heavy load lifting	76.3	69.8	84.6	<0.001
High job strain	65.2	58.4	73.9	<0.001
Low supervisor support	19.8	16.7	23.5	0.01
Low coworker support	22.8	18.5	28.4	0.001
Low job satisfaction	40.2	35.8	45.6	0.003
Mandatory overtime	8.5	11.4	4.8	0.001
Precarious work	22.7	21.5	24.3	0.4
Poor job security	17.4	16.9	18.1	0.7
Job instability	8.7	8.5	9.1	0.8
Smoking restriction policy	54.8	66.2	40.4	<0.001
Food access near workplace	53.0	60.8	42.9	<0.001
Designated eating areas	27.2	42.7	7.6	<0.001
	mean (SD)			
Supervisor support for safety	70.22 (±19.84)	71.74 (±19.04)	68.38 (±20.63)	0.01
Coworker support for safety	67.85 (±20.26)	70.03 (±18.51)	65.05 (±22.02)	<0.001
Commute time (minutes)	44.61 (±33.08)	46.54 (±39.31)	42.14 (±22.60)	0.03
*Health behaviors*	%			
Current cigarette smoker	27.3	24.9	30.5	0.07
Current e-cigarette user	10.7	10.0	11.5	0.5
	mean (SD)			
Alcohol consumption days/month	8.49 (±9.13)	7.96 (±8.61)	9.17 (±9.73)	0.05
Binge drinking days/month	4.28 (±6.11)	3.97 (±5.75)	4.67 (±6.52)	0.08
*Health outcomes*	%			
Neck/shoulder pain/discomfort	48.5	43.4	55.0	0.001
Hand/wrist pain/discomfort	53.5	46.4	62.6	<0.001
Lower back pain/discomfort	62.9	56.4	71.1	<0.001
Knee pain/discomfort	49.2	45.3	54.4	0.009
Doctor visit due to MS symptoms	22.0	19.3	25.6	0.03
Prescribed medication for pain	12.1	9.8	15.1	0.02
Poor physical health	25.0	21.7	29.0	0.01
Poor mental health	25.0	21.7	29.3	0.01
Low energy	39.7	36.3	44.1	0.02
Tired after work	55.1	49.9	61.5	0.001
*Work outcomes*	%			
Missed days—work-related injury	9.5	6.4	13.5	<0.001
Missed days—any injury or pain	20.5	16.6	25.4	0.002
Missed days due to any illness	41.3	42.0	40.6	0.7
Poor work ability	24.1	21.7	27.1	0.08
Poor health related productivity	14.8	11.8	18.5	0.007

MS = musculoskeletal. Note: Categorical variables displayed as *n* (%), continuous as mean (SD). ^a^ Total possible in each group, individual items may be less due to missing values. ^b^ Chi square and Wilcoxon Rank for categorical variables; *t*-test for continuous variables.

**Table 3 ijerph-18-08899-t003:** Univariate and multivariate associations between work outcomes, health outcomes, health behaviors, and exposure to work factors; *n* = 929; Prevalence Ratio (PR) and 95% Confidence Interval (CI) were calculated by using Poisson regression, with the same variables included in each model.

			Univariate		Multivariate	
Variable	*n*	Cases (%)	PR	95% CI	PR	95% CI
*Missed days due to work-related injury*						
Residential construction work	408	5.92	**2.12**	**(1.39–3.21)**	**1.89**	**(1.10–3.26)**
Mandatory overtime	73	0.49	0.61	(0.23–1.62)	0.41	(0.11–1.56)
Smoking restriction policy	509	4.69	0.81	(0.54–1.21)	0.97	(0.59–1.57)
Glove requirement policy	429	3.35	0.64	(0.42–0.98)	0.98	(0.54–1.76)
Frequent heavy load lifting	709	8.26	**2.09**	**(1.13–3.85)**	1.72	(0.86–3.44)
High job strain	563	7.56	**2.26**	**(1.31–3.89)**	**1.83**	**(1.03–3.26)**
Low coworker support	208	2.16	1.08	(0.66–1.76)	0.70	(0.39–1.24)
Low supervisor support	182	2.59	**1.57**	**(1.00–2.46)**	1.60	(0.94–2.74)
*Poor health related productivity*						
Residential construction work	408	8.16	**1.57**	**(1.14–2.15)**	**1.52**	**(1.01–2.30)**
Mandatory overtime	73	2.06	**1.91**	**(1.22–2.99)**	**1.67**	**(1.00–2.79)**
Smoking restriction policy	509	8.82	1.23	(0.89–1.69)	1.34	(0.93–1.94)
Glove requirement policy	429	6.14	0.83	(0.60–1.14)	0.82	(0.54–1.26)
Frequent heavy load lifting	709	12.39	**1.55**	**(1.01–2.39)**	1.60	(0.95–2.69)
High job strain	563	11.03	**1.81**	**(1.21–2.71)**	1.37	(0.88–2.14)
Low coworker support	208	5.22	**1.81**	**(1.31–2.50)**	**1.49**	**(1.02–2.19)**
Low supervisor support	182	4.74	**1.85**	**(1.34–2.57)**	1.19	(0.80–1.78)
*Lower back pain*/*discomfort*						
Residential construction work	408	31.09	**1.26**	**(1.14–1.39)**	**1.15**	**(1.02–1.30)**
Mandatory overtime	73	5.84	1.09	(0.93–1.29)	1.10	(0.92–1.32)
Smoking restriction policy	509	32.49	**0.88**	**(0.80–0.97)**	0.92	(0.82–1.03)
Glove requirement policy	429	26.78	**0.85**	**(0.77–0.94)**	0.98	(0.85–1.11)
Frequent heavy load lifting	709	51.70	**1.45**	**(1.25–1.69)**	**1.39**	**(1.18–1.64)**
High job strain	563	43.21	**1.21**	**(1.08–1.37)**	1.09	(0.96–1.24)
Low coworker support	208	16.01	**1.15**	**(1.03–1.28)**	1.08	(0.95–1.23)
Low supervisor support	182	14.20	**1.17**	**(1.05–1.31)**	1.07	(0.93–1.22)
*Tired after work*						
Residential construction work	408	27.08	**1.23**	**(1.10–1.38)**	**1.15**	**(1.00–1.33)**
Mandatory overtime	73	4.92	1.08	(0.88–1.32)	1.08	(0.89–1.33)
Smoking restriction policy	509	30.39	1.02	(0.91–1.14)	1.09	(0.96–1.25)
Glove requirement policy	429	24.14	0.91	(0.81–1.03)	1.02	(0.88–1.18)
Frequent heavy load lifting	709	46.55	**1.70**	**(1.41–2.05)**	**1.71**	**(1.38–2.11)**
High job strain	563	39.56	**1.35**	**(1.17–1.55)**	**1.23**	**(1.05–1.43)**
Low coworker support	208	14.29	**1.19**	**(1.05–1.35)**	1.11	(0.95–1.28)
Low supervisor support	182	12.75	**1.22**	**(1.08–1.39)**	1.12	(0.97–1.30)

Note: Bolded values show significant associations between independent variable and each outcome variable.

## Data Availability

The data presented in this study may be made available on request from the corresponding author, depending upon the nature of the request. The data are not publicly available due to ethical concerns and agreements with the participants.

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
