# Peer review of "Work Organization Factors Associated with Health and Work Outcomes among Apprentice Construction Workers: Comparison between the Residential and Commercial Sectors"

_ijerph, 2021, doi:10.3390/ijerph18178899_

Round 1
Reviewer 1 Report
Thank you very much for your excellent paper. Just a few comments.
I question the need to exclude women from the analysis since you don't know that there are "gendered" effects -- which might be determined with a sensitivity analysis. You still had reasonable numbers, if a low percentage, of women and I think it's important to include them where possible, otherwise the results do not represent the worker population -- or provide better reasons for excluding them.
5 line 148 – this line might be changed to clarify that these are worker-reported policies and perceived organization etc because you didn’t look at the policies or evaluate the work environment.
The reader really wants to know what determines who gets to work commercial vs. residential.
In the excellent background you discuss many reasons for why residential has worse conditions than commercial – which ones seemed applicable based on your results? Also, if you have any union staff perceptions of the reason for the differences, they would be great to include. I think that St. Louis is really different than almost anywhere in the country because there is union representation in the residential sector which is virtually unheard of elsewhere. I would think you would want to reflect more on the differences between union and non-union construction – maybe residential is the same everywhere, but what does that say about what the union is or isn’t doing to represent and advocate for their members on residential sites? Also, are these small residential sites or big residential developments? – which are characterized as commercial where I come from.
I would think you would want to interpret the potential differences between what is reported by these Apprentices who are generally young and inexperienced compared to the general construction worker population.
The last sentence of the paper is pretty opaque and I think it should be re-written. It's hard to make recommendations when you don't know why there are differences, only that there are, but I think you could come up with some.
Author Response
Reviewer 1
I question the need to exclude women from the analysis since you don't know that there are "gendered" effects -- which might be determined with a sensitivity analysis. You still had reasonable numbers, if a low percentage, of women and I think it's important to include them where possible, otherwise the results do not represent the worker population -- or provide better reasons for excluding them.
Response: We agree with the reviewer that including women in the analysis would provide a more real world view of the association between the female sex and a negative health outcome in the construction industry. Yet using a very small sample size may provide inaccurate results and could potentially lead to an incorrect conclusion. Statistically, a small sample size of only a few percent must have a very strong effect in order to show a meaningful association to an outcome, which is not likely. Yet, if we had oversampled on females to have a larger sample, and stratified the analysis, we may have found a meaningful association. Without the larger sample of women, we just don’t know if including the current sample size would give accurate results, so we chose to drop the small number of female cases.
5 line 148 – this line might be changed to clarify that these are worker-reported policies and perceived organization etc because you didn’t look at the policies or evaluate the work environment.
Response: We have made the recommended change to line 202.
The reader really wants to know what determines who gets to work commercial vs. residential.
Response: We have added more detail on lines 147-149 to explain that the type of work is linked to the contractor, and each apprentice has signed a letter of agreement to work for a signatory contractor. ”Note that each apprentice signs a letter of agreement to work for a signatory contractor. These contractors typically perform either residential construction (home building) or commercial construction.”
In the excellent background you discuss many reasons for why residential has worse conditions than commercial – which ones seemed applicable based on your results? Also, if you have any union staff perceptions of the reason for the differences, they would be great to include. I think that St. Louis is really different than almost anywhere in the country because there is union representation in the residential sector which is virtually unheard of elsewhere. I would think you would want to reflect more on the differences between union and non-union construction – maybe residential is the same everywhere, but what does that say about what the union is or isn’t doing to represent and advocate for their members on residential sites? Also, are these small residential sites or big residential developments? – which are characterized as commercial where I come from.
Response: We discussed how our findings were consistent with other studies with regard to having fewer workplace policies and poorer safety climates among residential apprentices in lines 380-389. We have added a sentence to point out that although these residential union apprentices have fewer safety policies in the workplace, residential work is more often performed by non-union workers who have less safety training than union workers (lines 402-404).
I would think you would want to interpret the potential differences between what is reported by these Apprentices who are generally young and inexperienced compared to the general construction worker population.
Response: We agree and have added a sentence in the limitations section (lines 425-427).
The last sentence of the paper is pretty opaque and I think it should be re-written. It's hard to make recommendations when you don't know why there are differences, only that there are, but I think you could come up with some.
Response: We have rewritten the last paragraph to more pointedly provide recommendations to address safety and health concerns in the residential sector as well as for the construction industry overall.
Reviewer 2 Report
Overall, this study and manuscript centered on work organization factors and health and work outcomes among construction workers is really sound and well-written. The authors have done a good job. The reviewer has some minor recommendations/suggestions below for further strengthening the manuscript.
- The reviewer would recommend a paragraph or two to begin the manuscript that focuses on work organization more broadly and the different populations that it has been studied in (healthcare workers, transportation, police officers/firefighters, farmworkers, white collar, etc.). There are some similarities but there are also differences. The authors could provide some context and then move into the focus on construction workers.
- In the methods/analysis, the reviewer would recommend providing more clarification on why certain work and health outcomes were selected. It just needs more description there and rationale for the decision-making.
- The reviewer would recommend a paragraph or two in the discussion/conclusions that focuses on how the results could practically be used in the workplace by employers or more broadly by policymakers.
Author Response
Overall, this study and manuscript centered on work organization factors and health and work outcomes among construction workers is really sound and well-written. The authors have done a good job. The reviewer has some minor recommendations/suggestions below for further strengthening the manuscript.
- The reviewer would recommend a paragraph or two to begin the manuscript that focuses on work organization more broadly and the different populations that it has been studied in (healthcare workers, transportation, police officers/firefighters, farmworkers, white collar, etc.). There are some similarities but there are also differences. The authors could provide some context and then move into the focus on construction workers.
Response: We have added a new introductory paragraph that discusses the problem of work organization and poor health outcomes (lines 36-41)
- In the methods/analysis, the reviewer would recommend providing more clarification on why certain work and health outcomes were selected. It just needs more description there and rationale for the decision-making.
Response: We have provided a brief explanation in lines 132-142 explaining that selection of the items of interest were primarily based on the higher prevalence of the factors and behaviors among construction workers from the four domains and many items that have shown an association between work and health outcomes.
- The reviewer would recommend a paragraph or two in the discussion/conclusions that focuses on how the results could practically be used in the workplace by employers or more broadly by policymakers.
Response: We have provided a clearer explanation in lines 355-368 of the conclusion about how the results may be applied to change worker health among the residential workforce as well as across the construction industry.
Reviewer 3 Report
This is a very good paper, well worth publishing, provided minor changes are made. See detailed comments in attached document.

Author Response
Review report
«Work Organization Factors Associated with Health and Work Outcomes among Apprentice Construction Workers: Comparison between the Residential and Commercial Sectors»
This paper has all the attributes of a publishable article. The introduction and clear and goes direct to the point, the objectives and hypotheses are well described, the methodology is correct and clearly presented, the results are presented in a state-of- the-art manner, the tables and figures are clear, the discussion is substantial, the authors are well aware of the limitations of their research, and the conclusions are in line with and supported by the results. The quality of the language is impeccable.
However, there are a few minor points which the authors might want to take into consideration in order to improve the article:
- The meaning at the bottom of Table 3 is unclear;
Response: We have made the requested revision.
- In line there is an unnecessary underscore (_);
Response: We were unable to find the unnecessary underscore (no line was indicated and a search of the document did not find the symbol intext).
- In line 325, there is a misplaced “s” (should read “outcomes and…”);
Response: We have made the requested revision.
- The conclusion is a bit short; and precisely because I agree that “the changing nature of work… will likely create poorer worker health…”, the authors should be bolder in proposing recommendations to address situations they have found in their study and to attenuate the effects of the changes they
Response: We have made substantial changes to the conclusion paragraph (lines 429-446).